# Deep Reinforcement Learning
# from Human Preferences

**Paul F Christiano**
OpenAI
paul@openai.com

**Jan Leike**
DeepMind
leike@google.com

**Tom B Brown**
Google Brain*
tombbrown@google.com

**Miljan Martic**
DeepMind
miljanm@google.com

**Shane Legg**
DeepMind
legg@google.com

**Dario Amodei**
OpenAI
damodei@openai.com

## Abstract

For sophisticated reinforcement learning (RL) systems to interact usefully with real-world environments, we need to communicate complex goals to these systems. In this work, we explore goals defined in terms of (non-expert) human preferences between pairs of trajectory segments. We show that this approach can effectively solve complex RL tasks without access to the reward function, including Atari games and simulated robot locomotion, while providing feedback on less than 1% of our agent's interactions with the environment. This reduces the cost of human oversight far enough that it can be practically applied to state-of-the-art RL systems. To demonstrate the flexibility of our approach, we show that we can successfully train complex novel behaviors with about an hour of human time. These behaviors and environments are considerably more complex than any which have been previously learned from human feedback.

## 1 Introduction

Recent success in scaling reinforcement learning (RL) to large problems has been driven in domains that have a well-specified reward function (Mnih et al., 2015, 2016; Silver et al., 2016). Unfortunately, many tasks involve goals that are complex, poorly-defined, or hard to specify. Overcoming this limitation would greatly expand the possible impact of deep RL and could increase the reach of machine learning more broadly.

For example, suppose that we wanted to use reinforcement learning to train a robot to clean a table or scramble an egg. It's not clear how to construct a suitable reward function, which will need to be a function of the robot's sensors. We could try to design a simple reward function that approximately captures the intended behavior, but this will often result in behavior that optimizes our reward function without actually satisfying our preferences. This difficulty underlies recent concerns about misalignment between our values and the objectives of our RL systems (Bostrom, 2014; Russell, 2016; Amodei et al., 2016). If we could successfully communicate our actual objectives to our agents, it would be a significant step towards addressing these concerns.

If we have demonstrations of the desired task, we can use inverse reinforcement learning (Ng and Russell, 2000) or imitation learning to copy the demonstrated behavior. But these approaches are not directly applicable to behaviors that are difficult for humans to demonstrate (such as controlling a robot with many degrees of freedom but non-human morphology).

An alternative approach is to allow a human to provide feedback on our system's current behavior and to use this feedback to define the task. In principle this fits within the paradigm of reinforcement learning, but using human feedback directly as a reward function is prohibitively expensive for RL systems that require hundreds or thousands of hours of experience. In order to practically train deep RL systems with human feedback, we need to decrease the amount of feedback required by several orders of magnitude.

We overcome this difficulty by asking humans to compare possible trajectories of the agent, using that data to learn a reward function, and optimizing the learned reward function with RL.

This basic approach has been explored in the past, but we confront the challenges involved in scaling it up to modern deep RL and demonstrate by far the most complex behaviors yet learned from human feedback.

Our experiments take place in two domains: Atari games in the Arcade Learning Environment (Bellemare et al., 2013), and robotics tasks in the physics simulator MuJoCo (Todorov et al., 2012). We show that a small amount of feedback from a non-expert human, ranging from fifteen minutes to five hours, suffice to learn both standard RL tasks and novel hard-to-specify behaviors such as performing a backflip or driving with the flow of traffic.

## 1.1 Related Work

A long line of work studies reinforcement learning from human ratings or rankings, including Akrour et al. (2011), Pilarski et al. (2011), Akrour et al. (2012), Wilson et al. (2012), Sugiyama et al. (2012), Wirth and Fürnkranz (2013), Daniel et al. (2015), El Asri et al. (2016), Wang et al. (2016), and Wirth et al. (2016). Other lines of research consider the general problem of reinforcement learning from preferences rather than absolute reward values (Fürnkranz et al., 2012; Akrour et al., 2014; Wirth et al., 2016), and optimizing using human preferences in settings other than reinforcement learning (Machwe and Parmee, 2006; Secretan et al., 2008; Brochu et al., 2010; Sørensen et al., 2016).

Our algorithm follows the same basic approach as Akrour et al. (2012) and Akrour et al. (2014), but considers much more complex domains and behaviors. The complexity of our environments force us to use different RL algorithms, reward models, and training strategies. One notable difference is that Akrour et al. (2012) and Akrour et al. (2014) elicit preferences over whole trajectories rather than short clips, and so would require about an order of magnitude more human time per data point. Our approach to feedback elicitation closely follows Wilson et al. (2012). However, Wilson et al. (2012) assumes that the reward function is the distance to some unknown (linear) "target" policy, and is never tested with real human feedback.

TAMER (Knox, 2012; Knox and Stone, 2013) also learns a reward function from human feedback, but learns from ratings rather than comparisons, has the human observe the agent as it behaves, and has been applied to settings where the desired policy can be learned orders of magnitude more quickly.

Compared to all prior work, our key contribution is to scale human feedback up to deep reinforcement learning and to learn much more complex behaviors. This fits into a recent trend of scaling reward learning methods to large deep learning systems, for example inverse RL (Finn et al., 2016), imitation learning (Ho and Ermon, 2016; Stadie et al., 2017), semi-supervised skill generalization (Finn et al., 2017), and bootstrapping RL from demonstrations (Silver et al., 2016; Hester et al., 2017).

## 2 Preliminaries and Method

### 2.1 Setting and Goal

We consider an agent interacting with an environment over a sequence of steps; at each time $t$ the agent receives an observation $o_t \in \mathcal{O}$ from the environment and then sends an action $a_t \in \mathcal{A}$ to the environment.

In traditional reinforcement learning, the environment would also supply a reward $r_t \in \mathbb{R}$ and the agent's goal would be to maximize the discounted sum of rewards. Instead of assuming that the environment produces a reward signal, we assume that there is a human overseer who can express

preferences between *trajectory segments*. A trajectory segment is a sequence of observations and actions, $\sigma = ((o_0, a_0), (o_1, a_1), \ldots, (o_{k-1}, a_{k-1})) \in (\mathcal{O} \times \mathcal{A})^k$. Write $\sigma^1 \succ \sigma^2$ to indicate that the human preferred trajectory segment $\sigma^1$ to trajectory segment $\sigma^2$. Informally, the goal of the agent is to produce trajectories which are preferred by the human, while making as few queries as possible to the human.

More precisely, we will evaluate our algorithms' behavior in two ways:

**Quantitative:** We say that preferences $\succ$ are *generated by* a reward function[2] $r : \mathcal{O} \times \mathcal{A} \to \mathbb{R}$ if

$$\left((o_0^1, a_0^1), \ldots, (o_{k-1}^1, a_{k-1}^1)\right) \succ \left((o_0^2, a_0^2), \ldots, (o_{k-1}^2, a_{k-1}^2)\right)$$

whenever

$$r\left(o_0^1, a_0^1\right) + \cdots + r\left(o_{k-1}^1, a_{k-1}^1\right) > r\left(o_0^2, a_0^2\right) + \cdots + r\left(o_{k-1}^2, a_{k-1}^2\right).$$

If the human's preferences are generated by a reward function $r$, then our agent ought to receive a high total reward according to $r$. So if we know the reward function $r$, we can evaluate the agent quantitatively. Ideally the agent will achieve reward nearly as high as if it had been using RL to optimize $r$.

**Qualitative:** Sometimes we have no reward function by which we can quantitatively evaluate behavior (this is the situation where our approach would be practically useful). In these cases, all we can do is qualitatively evaluate how well the agent satisfies the human's preferences. In this paper, we will start from a goal expressed in natural language, ask a human to evaluate the agent's behavior based on how well it fulfills that goal, and then present videos of agents attempting to fulfill that goal.

Our model based on trajectory segment comparisons is very similar to the trajectory preference queries used in Wilson et al. (2012), except that we don't assume that we can reset the system to an arbitrary state[3] and so our segments generally begin from different states. This complicates the interpretation of human comparisons, but we show that our algorithm overcomes this difficulty even when the human raters have no understanding of our algorithm.

## 2.2   Our Method

At each point in time our method maintains a policy $\pi : \mathcal{O} \to \mathcal{A}$ and a reward function estimate $\hat{r} : \mathcal{O} \times \mathcal{A} \to \mathbb{R}$, each parametrized by deep neural networks.

These networks are updated by three processes:

1. The policy $\pi$ interacts with the environment to produce a set of trajectories $\{\tau^1, \ldots, \tau^i\}$. The parameters of $\pi$ are updated by a traditional reinforcement learning algorithm, in order to maximize the sum of the predicted rewards $r_t = \hat{r}(o_t, a_t)$.

2. We select pairs of segments $(\sigma^1, \sigma^2)$ from the trajectories $\{\tau^1, \ldots, \tau^i\}$ produced in step 1, and send them to a human for comparison.

3. The parameters of the mapping $\hat{r}$ are optimized via supervised learning to fit the comparisons collected from the human so far.

These processes run asynchronously, with trajectories flowing from process (1) to process (2), human comparisons flowing from process (2) to process (3), and parameters for $\hat{r}$ flowing from process (3) to process (1). The following subsections provide details on each of these processes.

### 2.2.1 Optimizing the Policy

After using $\hat{r}$ to compute rewards, we are left with a traditional reinforcement learning problem. We can solve this problem using any RL algorithm that is appropriate for the domain. One subtlety is that the reward function $\hat{r}$ may be non-stationary, which leads us to prefer methods which are robust to changes in the reward function. This led us to focus on policy gradient methods, which have been applied successfully for such problems (Ho and Ermon, 2016).

In this paper, we use *advantage actor-critic* (A2C; Mnih et al., 2016) to play Atari games, and *trust region policy optimization* (TRPO; Schulman et al., 2015) to perform simulated robotics tasks. In each case, we used parameter settings which have been found to work well for traditional RL tasks. The only hyperparameter which we adjusted was the entropy bonus for TRPO. This is because TRPO relies on the trust region to ensure adequate exploration, which can lead to inadequate exploration if the reward function is changing.

We normalized the rewards produced by $\hat{r}$ to have zero mean and constant standard deviation. This is a typical preprocessing step which is particularly appropriate here since the position of the rewards is underdetermined by our learning problem.

### 2.2.2 Preference Elicitation

The human overseer is given a visualization of two trajectory segments, in the form of short movie clips. In all of our experiments, these clips are between 1 and 2 seconds long.

The human then indicates which segment they prefer, that the two segments are equally good, or that they are unable to compare the two segments.

The human judgments are recorded in a database $\mathcal{D}$ of triples $(\sigma^1, \sigma^2, \mu)$, where $\sigma^1$ and $\sigma^2$ are the two segments and $\mu$ is a distribution over $\{1, 2\}$ indicating which segment the user preferred. If the human selects one segment as preferable, then $\mu$ puts all of its mass on that choice. If the human marks the segments as equally preferable, then $\mu$ is uniform. Finally, if the human marks the segments as incomparable, then the comparison is not included in the database.

### 2.2.3 Fitting the Reward Function

We can interpret a reward function estimate $\hat{r}$ as a preference-predictor if we view $\hat{r}$ as a latent factor explaining the human's judgments and assume that the human's probability of preferring a segment $\sigma^i$ depends exponentially on the value of the latent reward summed over the length of the clip:[4]

$$\hat{P}\big[\sigma^1 \succ \sigma^2\big] = \frac{\exp \sum \hat{r}\big(o_t^1, a_t^1\big)}{\exp \sum \hat{r}(o_t^1, a_t^1) + \exp \sum \hat{r}(o_t^2, a_t^2)}. \tag{1}$$

We choose $\hat{r}$ to minimize the cross-entropy loss between these predictions and the actual human labels:

$$\text{loss}(\hat{r}) = - \sum_{(\sigma^1, \sigma^2, \mu) \in \mathcal{D}} \mu(1) \log \hat{P}\big[\sigma^1 \succ \sigma^2\big] + \mu(2) \log \hat{P}\big[\sigma^2 \succ \sigma^1\big].$$

This follows the Bradley-Terry model (Bradley and Terry, 1952) for estimating score functions from pairwise preferences, and is the specialization of the Luce-Shephard choice rule (Luce, 2005; Shepard, 1957) to preferences over trajectory segments.

Our actual algorithm incorporates a number of modifications to this basic approach, which early experiments discovered to be helpful and which are analyzed in Section 3.3:

- We fit an ensemble of predictors, each trained on $|\mathcal{D}|$ triples sampled from $\mathcal{D}$ with replacement. The estimate $\hat{r}$ is defined by independently normalizing each of these predictors and then averaging the results.
- A fraction of $1/e$ of the data is held out to be used as a validation set for each predictor. We use $\ell_2$ regularization and adjust the regularization coefficient to keep the validation loss between $1.1$ and $1.5$ times the training loss. In some domains we also apply dropout for regularization.

- Rather than applying a softmax directly as described in Equation 1, we assume there is a 10% chance that the human responds uniformly at random. Conceptually this adjustment is needed because human raters have a constant probability of making an error, which doesn't decay to 0 as the difference in reward difference becomes extreme.

### 2.2.4 Selecting Queries

We decide how to query preferences based on an approximation to the uncertainty in the reward function estimator, similar to Daniel et al. (2014): we sample a large number of pairs of trajectory segments of length $k$ from the latest agent-environment interactions, use each reward predictor in our ensemble to predict which segment will be preferred from each pair, and then select those trajectories for which the predictions have the highest variance across ensemble members[5] This is a crude approximation and the ablation experiments in Section 3 show that in some tasks it actually impairs performance. Ideally, we would want to query based on the expected value of information of the query (Akrour et al., 2012; Krueger et al., 2016), but we leave it to future work to explore this direction further.

## 3 Experimental Results

We implemented our algorithm in TensorFlow (Abadi et al., 2016). We interface with MuJoCo (Todorov et al., 2012) and the Arcade Learning Environment (Bellemare et al., 2013) through the OpenAI Gym (Brockman et al., 2016).

### 3.1 Reinforcement Learning Tasks with Unobserved Rewards

In our first set of experiments, we attempt to solve a range of benchmark tasks for deep RL *without observing the true reward*. Instead, the agent learns about the goal of the task only by asking a human which of two trajectory segments is better. Our goal is to solve the task in a reasonable amount of time using as few queries as possible.

In our experiments, feedback is provided by contractors who are given a 1-2 sentence description of each task before being asked to compare several hundred to several thousand pairs of trajectory segments for that task (see Appendix B for the exact instructions given to contractors). Each trajectory segment is between 1 and 2 seconds long. Contractors responded to the average query in 3-5 seconds, and so the experiments involving real human feedback required between 30 minutes and 5 hours of human time.

For comparison, we also run experiments using a synthetic oracle whose preferences are generated (in the sense of Section 2.1) by the real reward[6]. We also compare to the baseline of RL training using the real reward. Our aim here is not to outperform but rather to do nearly as well as RL without access to reward information and instead relying on much scarcer feedback. Nevertheless, note that feedback from real humans does have the potential to outperform RL (and as shown below it actually does so on some tasks), because the human feedback might provide a better-shaped reward.

We describe the details of our experiments in Appendix A, including model architectures, modifications to the environment, and the RL algorithms used to optimize the policy.

### 3.1.1 Simulated Robotics

The first tasks we consider are eight simulated robotics tasks, implemented in MuJoCo (Todorov et al., 2012), and included in OpenAI Gym (Brockman et al., 2016). We made small modifications to these tasks in order to avoid encoding information about the task in the environment itself (the modifications are described in detail in Appendix A). The reward functions in these tasks are quadratic functions of distances, positions and velocities, and most are linear. We included a simple cartpole

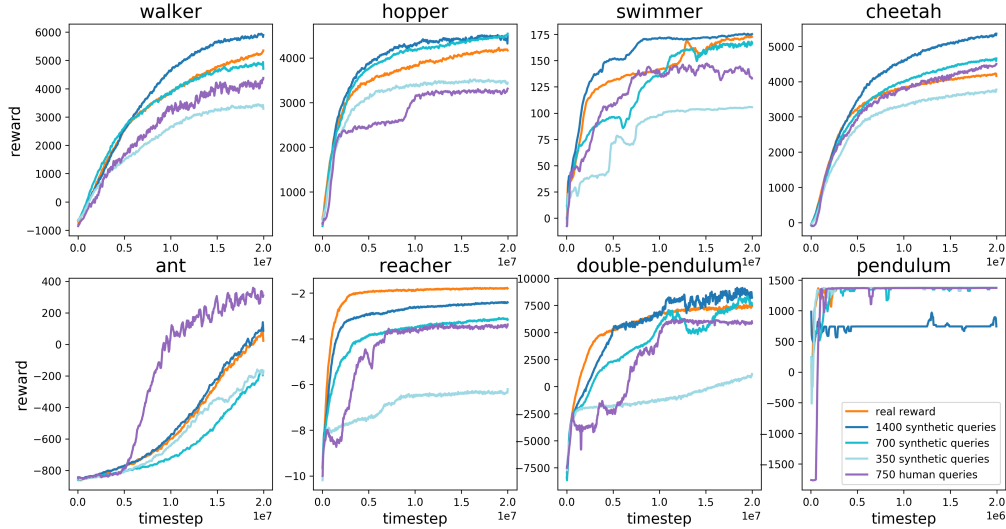

Figure 1: Results on MuJoCo simulated robotics as measured on the tasks' true reward. We compare our method using real human feedback (purple), our method using synthetic feedback provided by an oracle (shades of blue), and reinforcement learning using the true reward function (orange). All curves are the average of 5 runs, except for the real human feedback, which is a single run, and each point is the average reward over five consecutive batches. For Reacher and Cheetah feedback was provided by an author due to time constraints. For all other tasks, feedback was provided by contractors unfamiliar with the environments and with our algorithm. The irregular progress on Hopper is due to one contractor deviating from the typical labeling schedule.

task ("pendulum") for comparison, since this is representative of the complexity of tasks studied in prior work.

Figure 1 shows the results of training our agent with 700 queries to a human rater, compared to learning from 350, 700, or 1400 synthetic queries, as well as to RL learning from the real reward. With 700 labels we are able to nearly match reinforcement learning on all of these tasks. Training with learned reward functions tends to be less stable and higher variance, while having a comparable mean performance.

Surprisingly, by 1400 labels our algorithm performs slightly better than if it had simply been given the true reward, perhaps because the learned reward function is slightly better shaped—the reward learning procedure assigns positive rewards to all behaviors that are typically followed by high reward. The difference may also be due to subtle changes in the relative scale of rewards or our use of entropy regularization.

Real human feedback is typically only slightly less effective than the synthetic feedback; depending on the task human feedback ranged from being half as efficient as ground truth feedback to being equally efficient. On the Ant task the human feedback significantly outperformed the synthetic feedback, apparently because we asked humans to prefer trajectories where the robot was "standing upright," which proved to be useful reward shaping. (There was a similar bonus in the RL reward function to encourage the robot to remain upright, but the simple hand-crafted bonus was not as useful.)

### 3.1.2  Atari

The second set of tasks we consider is a set of seven Atari games in the Arcade Learning Environment (Bellemare et al., 2013), the same games presented in Mnih et al., 2013.

Figure 2 shows the results of training our agent with 5,500 queries to a human rater, compared to learning from 350, 700, or 1400 synthetic queries, as well as to RL learning from the real reward. Our method has more difficulty matching RL in these challenging environments, but nevertheless it displays substantial learning on most of them and matches or even exceeds RL on some. Specifically,

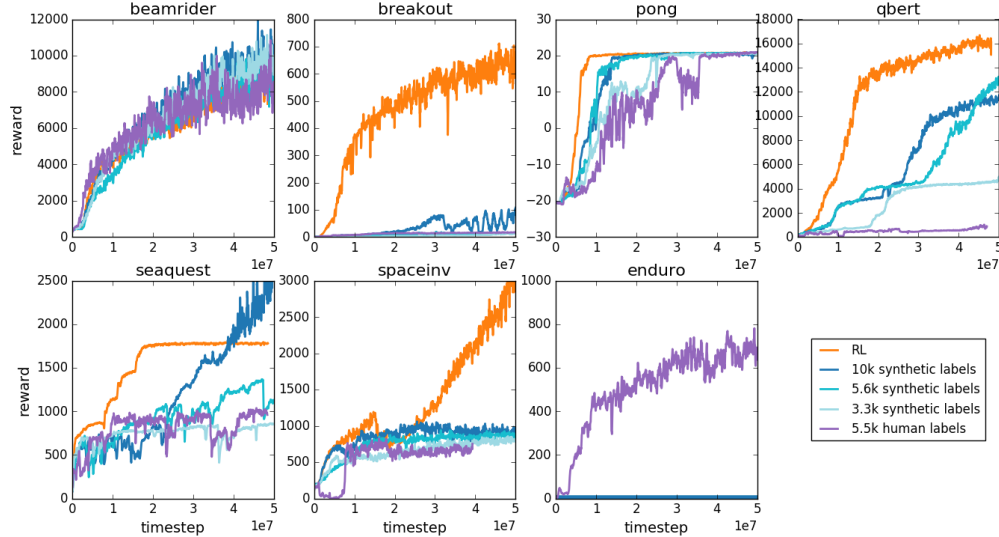

Figure 2: Results on Atari games as measured on the tasks' true reward. We compare our method using real human feedback (purple), our method using synthetic feedback provided by an oracle (shades of blue), and reinforcement learning using the true reward function (orange). All curves are the average of 3 runs, except for the real human feedback which is a single run, and each point is the average reward over about 150,000 consecutive frames.

on BeamRider and Pong, synthetic labels match or come close to RL even with only 3,300 such labels. On Seaquest and Qbert synthetic feedback eventually performs near the level of RL but learns more slowly. On SpaceInvaders and Breakout synthetic feedback never matches RL, but nevertheless the agent improves substantially, often passing the first level in SpaceInvaders and reaching a score of 20 on Breakout, or 50 with enough labels.

On most of the games real human feedback performs similar to or slightly worse than synthetic feedback with the same number of labels, and often comparably to synthetic feedback that has 40% fewer labels. On Qbert, our method fails to learn to beat the first level with real human feedback; this may be because short clips in Qbert can be confusing and difficult to evaluate. Finally, Enduro is difficult for A3C to learn due to the difficulty of successfully passing other cars through random exploration, and is correspondingly difficult to learn with synthetic labels, but human labelers tend to reward any progress towards passing cars, essentially shaping the reward and thus outperforming A3C in this game (the results are comparable to those achieved with DQN).

## 3.2 Novel behaviors

Experiments with traditional RL tasks help us understand whether our method is effective, but the ultimate purpose of human interaction is to solve tasks for which no reward function is available.

Using the same parameters as in the previous experiments, we show that our algorithm can learn novel complex behaviors. We demonstrate:

1. The Hopper robot performing a sequence of backflips (see Figure 4). This behavior was trained using 900 queries in less than an hour. The agent learns to consistently perform a backflip, land upright, and repeat.

2. The Half-Cheetah robot moving forward while standing on one leg. This behavior was trained using 800 queries in under an hour.

3. Keeping alongside other cars in Enduro. This was trained with roughly 1,300 queries and 4 million frames of interaction with the environment; the agent learns to stay almost exactly even with other moving cars for a substantial fraction of the episode, although it gets confused by changes in background.

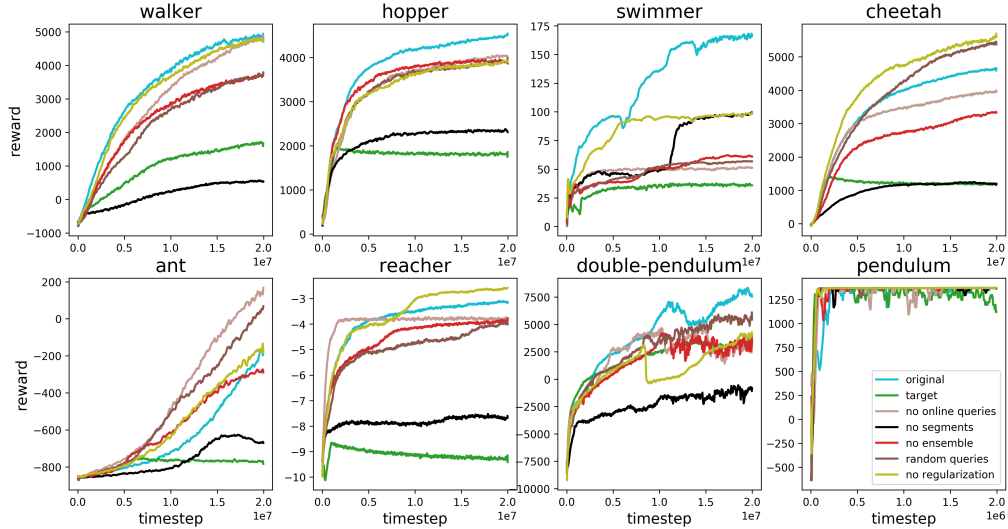

Figure 3: Performance of our algorithm on MuJoCo tasks after removing various components, as described in Section Section 3.3. All graphs are averaged over 5 runs, using 700 synthetic labels each.

Videos of these behaviors can be found at `https://goo.gl/MhgvIU`. These behaviors were trained using feedback from the authors.

### 3.3  Ablation Studies

In order to better understand the performance of our algorithm, we consider a range of modifications:

1. We pick queries uniformly at random rather than prioritizing queries for which there is disagreement (**random queries**).

2. We train only one predictor rather than an ensemble (**no ensemble**). In this setting, we also choose queries at random, since there is no longer an ensemble that we could use to estimate disagreement.

3. We train on queries only gathered at the beginning of training, rather than gathered throughout training (**no online queries**).

4. We remove the $\ell_2$ regularization and use only dropout (**no regularization**).

5. On the robotics tasks only, we use trajectory segments of length 1 (**no segments**).

6. Rather than fitting $\hat{r}$ using comparisons, we consider an oracle which provides the true total reward over a trajectory segment, and fit $\hat{r}$ to these total rewards using mean squared error (**target**).

The results are presented in Figure 3 for MuJoCo and Figure 4 for Atari.

Training the reward predictor offline can lead to bizarre behavior that is undesirable as measured by the true reward (Amodei et al., 2016). For instance, on Pong offline training sometimes leads our agent to avoid losing points but not to score points; this can result in extremely long volleys (videos at `https://goo.gl/L5eAbk`). This type of behavior demonstrates that in general human feedback needs to be intertwined with RL rather than provided statically.

Our main motivation for eliciting comparisons rather than absolute scores was that we found it much easier for humans to provide consistent comparisons than consistent absolute scores, especially on the continuous control tasks and on the qualitative tasks in Section 3.2; nevertheless it seems important to understand how using comparisons affects performance. For continuous control tasks we found that predicting comparisons worked much better than predicting scores. This is likely because the scale of rewards varies substantially and this complicates the regression problem, which is smoothed significantly when we only need to predict comparisons. In the Atari tasks we clipped rewards

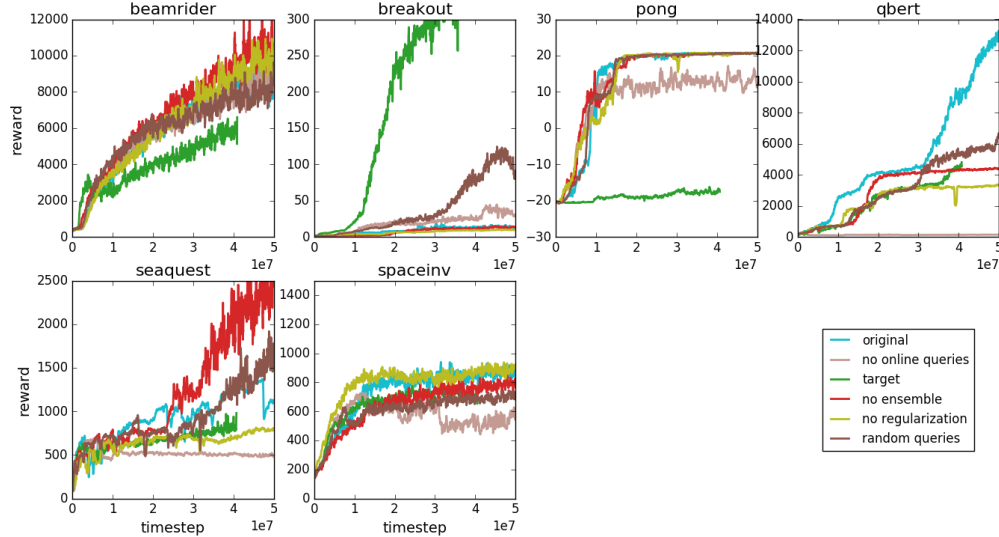

Figure 4: Performance of our algorithm on Atari tasks after removing various components, as described in Section 3.3. All curves are an average of 3 runs using 5,500 synthetic labels (see minor exceptions in Section A.2).

and effectively only predicted the sign, avoiding these difficulties (this is not a suitable solution for the continuous control tasks because the magnitude of the reward is important to learning). In these tasks comparisons and targets had significantly different performance, but neither consistently outperformed the other.

We also observed large performance differences when using single frames rather than clips.[7] In order to obtain the same results using single frames we would need to have collected significantly more comparisons. In general we discovered that asking humans to compare longer clips was significantly more helpful *per clip*, and significantly less helpful *per frame*. Shrinking the clip length below 1-2 seconds did not significantly decrease the human time required to label each clip in early experiments, and so seems less efficient *per second of human time*. In the Atari environments we also found that it was often easier to compare longer clips because they provide more context than single frames.

## 4  Discussion and Conclusions

Agent-environment interactions are often radically cheaper than human interaction. We show that by learning a separate reward model using supervised learning, it is possible to reduce the interaction complexity by roughly 3 orders of magnitude.

Although there is a large literature on preference elicitation and reinforcement learning from unknown reward functions, we provide the first evidence that these techniques can be economically scaled up to state-of-the-art reinforcement learning systems. This represents a step towards practical applications of deep RL to complex real-world tasks.

In the long run it would be desirable to make learning a task from human preferences no more difficult than learning it from a programmatic reward signal, ensuring that powerful RL systems can be applied in the service of complex human values rather than low-complexity goals.

### Acknowledgments

We thank Olivier Pietquin, Bilal Piot, Laurent Orseau, Pedro Ortega, Victoria Krakovna, Owain Evans, Andrej Karpathy, Igor Mordatch, and Jack Clark for reading drafts of the paper. We thank Tyler Adkisson, Mandy Beri, Jessica Richards, Heather Tran, and other contractors for providing the

## Footnotes

*Work done while at OpenAI.

[2]Here we assume here that the reward is a function of the observation and action. In our experiments in Atari environments, we instead assume the reward is a function of the preceding 4 observations. In a general partially observable environment, we could instead consider reward functions that depend on the whole sequence of observations, and model this reward function with a recurrent neural network.

[3]Wilson et al. (2012) also assumes the ability to sample reasonable initial states. But we work with high dimensional state spaces for which random states will not be reachable and the intended policy inhabits a low-dimensional manifold.

[4]Equation 1 does not use discounting, which could be interpreted as modeling the human to be indifferent about when things happen in the trajectory segment. Using explicit discounting or inferring the human's discount function would also be reasonable choices.

[5]Note that trajectory segments almost never start from the same state.

[6]In the case of Atari games with sparse rewards, it is relatively common for two clips to both have zero reward in which case the oracle outputs indifference. Because we considered clips rather than individual states, such ties never made up a large majority of our data. Moreover, ties still provide significant information to the reward predictor as long as they are not too common.

[7]We only ran these tests on continuous control tasks because our Atari reward model depends on a sequence of consecutive frames rather than a single frame, as described in Section A.2
