[Supplementary Material]

data used to train our agents. Finally, we thank OpenAI and DeepMind for providing a supportive research environment and for supporting and encouraging this collaboration.

## Footnotes

[8] All of these reward functions are second degree polynomials of the input features, and so if we were concerned only with these tasks we could take a simpler approach to learning the reward function. However, using this more flexible architecture allows us to immediately generalize to tasks for which the reward function is not so simple, as described in Section 3.2.

[9]e.g. http://www.free80sarcade.com/2600_Beamrider.php

# References

Martin Abadi et al. Tensorflow: Large-scale machine learning on heterogeneous distributed systems. *arXiv preprint arXiv:1603.04467*, 2016.

Riad Akrour, Marc Schoenauer, and Michele Sebag. Preference-based policy learning. *Machine learning and knowledge discovery in databases*, pages 12–27, 2011.

Riad Akrour, Marc Schoenauer, and Michèle Sebag. April: Active preference learning-based reinforcement learning. In *Joint European Conference on Machine Learning and Knowledge Discovery in Databases*, pages 116–131, 2012.

Riad Akrour, Marc Schoenauer, Michèle Sebag, and Jean-Christophe Souplet. Programming by feedback. In *International Conference on Machine Learning*, pages 1503–1511, 2014.

Dario Amodei, Chris Olah, Jacob Steinhardt, Paul Christiano, John Schulman, and Dan Mané. Concrete problems in AI safety. *arXiv preprint arXiv:1606.06565*, 2016.

Marc G Bellemare, Yavar Naddaf, Joel Veness, and Michael Bowling. The Arcade Learning Environment: An evaluation platform for general agents. *Journal of Artificial Intelligence Research*, 47:253–279, 2013.

Nick Bostrom. *Superintelligence: Paths, Dangers, Strategies*. Oxford University Press, 2014.

Ralph Allan Bradley and Milton E Terry. Rank analysis of incomplete block designs: I. The method of paired comparisons. *Biometrika*, 39(3/4):324–345, 1952.

Eric Brochu, Tyson Brochu, and Nando de Freitas. A bayesian interactive optimization approach to procedural animation design. In *Proceedings of the 2010 ACM SIGGRAPH/Eurographics Symposium on Computer Animation*, pages 103–112. Eurographics Association, 2010.

Greg Brockman, Vicki Cheung, Ludwig Pettersson, Jonas Schneider, John Schulman, Jie Tang, and Wojciech Zaremba. OpenAI Gym. *arXiv preprint arXiv:1606.01540*, 2016.

Christian Daniel, Malte Viering, Jan Metz, Oliver Kroemer, and Jan Peters. Active reward learning. In *Robotics: Science and Systems*, 2014.

Christian Daniel, Oliver Kroemer, Malte Viering, Jan Metz, and Jan Peters. Active reward learning with a novel acquisition function. *Autonomous Robots*, 39(3):389–405, 2015.

Layla El Asri, Bilal Piot, Matthieu Geist, Romain Laroche, and Olivier Pietquin. Score-based inverse reinforcement learning. In *International Conference on Autonomous Agents and Multiagent Systems*, pages 457–465, 2016.

Chelsea Finn, Sergey Levine, and Pieter Abbeel. Guided cost learning: Deep inverse optimal control via policy optimization. In *International Conference on Machine Learning*, volume 48, 2016.

Chelsea Finn, Tianhe Yu, Justin Fu, Pieter Abbeel, and Sergey Levine. Generalizing skills with semi-supervised reinforcement learning. In *International Conference on Learning Representations*, 2017.

Johannes Fürnkranz, Eyke Hüllermeier, Weiwei Cheng, and Sang-Hyeun Park. Preference-based reinforcement learning: A formal framework and a policy iteration algorithm. *Machine learning*, 89(1-2):123–156, 2012.

Todd Hester, Matej Vecerik, Olivier Pietquin, Marc Lanctot, Tom Schaul, Bilal Piot, Andrew Sendonaris, Gabriel Dulac-Arnold, Ian Osband, John Agapiou, Joel Z Leibo, and Audrunas Gruslys. Learning from demonstrations for real world reinforcement learning. *arXiv preprint arXiv:1704.03732*, 2017.

Jonathan Ho and Stefano Ermon. Generative adversarial imitation learning. In *Advances in Neural Information Processing Systems*, pages 4565–4573, 2016.

W. Bradley Knox and Peter Stone. Learning non-myopically from human-generated reward. In *Intelligent User Interfaces*, pages 191–202, 2013.

William Bradley Knox. *Learning from human-generated reward*. PhD thesis, University of Texas at Austin, 2012.

David Krueger, Jan Leike, Owain Evans, and John Salvatier. Active reinforcement learning: Observing rewards at a cost. In *Future of Interactive Learning Machines, NIPS Workshop*, 2016.

R Duncan Luce. *Individual choice behavior: A theoretical analysis*. Courier Corporation, 2005.

AT Machwe and IC Parmee. Introducing machine learning within an interactive evolutionary design environment. In *DS 36: Proceedings DESIGN 2006, the 9th International Design Conference, Dubrovnik, Croatia*, 2006.

Volodymyr Mnih, Koray Kavukcuoglu, David Silver, Alex Graves, Ioannis Antonoglou, Daan Wierstra, and Martin Riedmiller. Playing Atari with deep reinforcement learning. *arXiv preprint arXiv:1312.5602*, 2013.

Volodymyr Mnih, Koray Kavukcuoglu, David Silver, Andrei A Rusu, Joel Veness, Marc G Bellemare, Alex Graves, Martin Riedmiller, Andreas K Fidjeland, Georg Ostrovski, Stig Petersen, Charles Beattie, Amir Sadik, Ioannis Antonoglou, Helen King, Dharshan Kumaran, Daan Wierstra, Shane Legg, and Demis Hassabis. Human-level control through deep reinforcement learning. *Nature*, 518(7540):529–533, 2015.

Volodymyr Mnih, Adria Puigdomenech Badia, Mehdi Mirza, Alex Graves, Timothy Lillicrap, Tim Harley, David Silver, and Koray Kavukcuoglu. Asynchronous methods for deep reinforcement learning. In *International Conference on Machine Learning*, pages 1928–1937, 2016.

Andrew Y Ng and Stuart Russell. Algorithms for inverse reinforcement learning. In *International Conference on Machine learning*, pages 663–670, 2000.

Patrick M Pilarski, Michael R Dawson, Thomas Degris, Farbod Fahimi, Jason P Carey, and Richard Sutton. Online human training of a myoelectric prosthesis controller via actor-critic reinforcement learning. In *International Conference on Rehabilitation Robotics*, pages 1–7, 2011.

Stuart Russell. Should we fear supersmart robots? *Scientific American*, 314(6):58, 2016.

John Schulman, Sergey Levine, Pieter Abbeel, Michael I Jordan, and Philipp Moritz. Trust region policy optimization. In *International Conference on Machine Learning*, pages 1889–1897, 2015.

Jimmy Secretan, Nicholas Beato, David B D Ambrosio, Adelein Rodriguez, Adam Campbell, and Kenneth O Stanley. Picbreeder: Evolving pictures collaboratively online. In *Conference on Human Factors in Computing Systems*, pages 1759–1768, 2008.

Roger N Shepard. Stimulus and response generalization: A stochastic model relating generalization to distance in psychological space. *Psychometrika*, 22(4):325–345, 1957.

David Silver, Aja Huang, Chris J Maddison, Arthur Guez, Laurent Sifre, George Van Den Driessche, Julian Schrittwieser, Ioannis Antonoglou, Veda Panneershelvam, Marc Lanctot, Sander Dieleman, Dominik Grewe, John Nham, Nal Kalchbrenner, Ilya Sutskever, Timothy Lillicrap, Madeleine Leach, Koray Kavukcuoglu, Thore Graepel, and Demis Hassabis. Mastering the game of Go with deep neural networks and tree search. *Nature*, 529(7587):484–489, 2016.

Patrikk D Sørensen, Jeppeh M Olsen, and Sebastian Risi. Breeding a diversity of super mario behaviors through interactive evolution. In *Computational Intelligence and Games (CIG), 2016 IEEE Conference on*, pages 1–7. IEEE, 2016.

Bradly C Stadie, Pieter Abbeel, and Ilya Sutskever. Third-person imitation learning. In *International Conference on Learning Representations*, 2017.

Hiroaki Sugiyama, Toyomi Meguro, and Yasuhiro Minami. Preference-learning based inverse reinforcement learning for dialog control. In *INTERSPEECH*, pages 222–225, 2012.

Emanuel Todorov, Tom Erez, and Yuval Tassa. Mujoco: A physics engine for model-based control. In *International Conference on Intelligent Robots and Systems*, pages 5026–5033, 2012.

Sida I Wang, Percy Liang, and Christopher D Manning. Learning language games through interaction. *arXiv preprint arXiv:1606.02447*, 2016.

Aaron Wilson, Alan Fern, and Prasad Tadepalli. A Bayesian approach for policy learning from trajectory preference queries. In *Advances in Neural Information Processing Systems*, pages 1133–1141, 2012.

Christian Wirth and Johannes Fürnkranz. Preference-based reinforcement learning: A preliminary survey. In *ECML/PKDD Workshop on Reinforcement Learning from Generalized Feedback: Beyond Numeric Rewards*, 2013.

Christian Wirth, J Fürnkranz, Gerhard Neumann, et al. Model-free preference-based reinforcement learning. In *AAAI*, pages 2222–2228, 2016.

# A   Experimental Details

Many RL environments have termination conditions that depend on the behavior of the agent, such as ending an episode when the agent dies or falls over. We found that such termination conditions encode information about the task even when the reward function is not observable. To avoid this subtle source of supervision, which could potentially confound our attempts to learn from human preferences only, we removed all variable-length episodes:

- In the Gym versions of our robotics tasks, the episode ends when certain parameters go outside of a prescribed range (for example when the robot falls over). We replaced these termination conditions by a penalty which encourages the parameters to remain in the range (and which the agent must learn).

- In Atari games, we do not send life loss or episode end signals to the agent (we do continue to actually reset the environment), effectively converting the environment into a single continuous episode. When providing synthetic oracle feedback we replace episode ends with a penalty in all games except Pong; the agent must learn this penalty.

Removing variable length episodes leaves the agent with only the information encoded in the environment itself; human feedback provides its only guidance about what it ought to do.

At the beginning of training we compare a number of trajectory segments drawn from rollouts of an untrained (randomly initialized) policy. In the Atari domain we also pretrain the reward predictor for 200 epochs before beginning RL training, to reduce the likelihood of irreversibly learning a bad policy based on an untrained predictor. For the rest of training, labels are fed in at a rate decaying inversely with the number of timesteps; after twice as many timesteps have elapsed, we answer about half as many queries per unit time. The details of this schedule are described in each section. This "label annealing" allows us to balance the importance of having a good predictor from the start with the need to adapt the predictor as the RL agent learns and encounters new states. When training with real human feedback, we attempt to similarly anneal the label rate, although in practice this is approximate because contractors give feedback at uneven rates.

Except where otherwise stated we use an ensemble of 3 predictors, and draw a factor 10 more clip pair candidates than we ultimately present to the human, with the presented clips being selected via maximum variance between the different predictors as described in Section 2.2.4.

## A.1   Simulated Robotics Tasks

The OpenAI Gym continuous control tasks penalize large torques. Because torques are not directly visible to a human supervisor, these reward functions are not good representatives of human preferences over trajectories and so we removed them.

For the simulated robotics tasks, we optimize policies using *trust region policy optimization* (TRPO, Schulman et al., 2015) with discount rate $\gamma = 0.995$ and $\lambda = 0.97$. The reward predictor is a two-layer neural network with 64 hidden units each, using leaky ReLUs ($\alpha = 0.01$) as nonlinearities.[8] We compare trajectory segments that last 1.5 seconds, which varies from 15 to 60 timesteps depending on the task.

We normalize the reward predictions to have standard deviation 1. When learning from the reward predictor, we add an entropy bonus of 0.01 on all tasks except swimmer, where we use an entropy bonus of 0.001. As noted in Section 2.2.1, this entropy bonus helps to incentivize the increased exploration needed to deal with a changing reward function.

We collect $25\%$ of our comparisons from a randomly initialized policy network at the beginning of training, and our rate of labeling after $T$ frames $2 * 10^6 / (T + 2 * 10^6)$.

## A.2 Atari

Our Atari agents are trained using the standard set of environment wrappers used by Mnih et al. (2015): 0 to 30 no-ops in the beginning of an episode, max-pooling over adjacent frames, stacking of 4 frames, a frameskip of 4, life loss ending an episode (but not resetting the environment), and rewards clipped to $[-1, 1]$.

Atari games include a visual display of the score, which in theory could be used to trivially infer the reward. Since we want to focus instead on inferring the reward from the complex dynamics happening in the game, we replace the score area with a constant black background on all seven games. On BeamRider we additionally blank out the enemy ship count, and on Enduro we blank out the speedometer.

For the Atari tasks we optimize policies using the A3C algorithm (Mnih et al., 2016) in synchronous form (A2C), with policy architecture as described in Mnih et al. (2015). We use standard settings for the hyperparameters: an entropy bonus of $\beta = 0.01$, learning rate of $0.0007$ decayed linearly to reach zero after 80 million timesteps (although runs were actually trained for only 50 million timesteps), $n = 5$ steps per update, $N = 16$ parallel workers, discount rate $\gamma = 0.99$, and policy gradient using Adam with $\alpha = 0.99$ and $\epsilon = 10^{-5}$.

For the reward predictor, we use 84x84 images as inputs (the same as the inputs to the policy), and stack 4 frames for a total 84x84x4 input tensor. This input is fed through 4 convolutional layers of size 7x7, 5x5, 3x3, and 3x3 with strides 3, 2, 1, 1, each having 16 filters, with leaky ReLU nonlinearities ($\alpha = 0.01$). This is followed by a fully connected layer of size 64 and then a scalar output. All convolutional layers use batch norm and dropout with $\alpha = 0.5$ to prevent predictor overfitting. In addition we use $\ell_2$ regularization with the adapative scheme described in Section 2.2.3. Since the reward predictor is ultimately used to compare two sums over timesteps, its scale is arbitrary, and we normalize it to have a standard deviation of 0.05 (we could equivalently have adjusted our learning rates and entropy bonus, but this choice allowed us to use the same parameters as for the real reward function).

We compare trajectory segments of 25 timesteps (1.7 seconds at 15 fps with frame skipping).

We collect 500 comparisons from a randomly initialized policy network at the beginning of training, and our rate of labeling after $T$ frames of training is decreased every $5 * 10^6$ frames, to be roughly proportional to $5 * 10^6/(T + 5 * 10^6)$.

The predictor is trained asynchronously from the RL agent, and on our hardware typically processes 1 label per 10 RL timesteps. We maintain a buffer of only the last 3,000 labels and loop over this buffer continuously; this is to ensure that the predictor gives enough weight to new labels (which can represent a shift in distribution) when the total number of labels becomes large.

In the ablation studies of Figure 5b, pretraining has 5,000 labels rather than 5,500, and the "target" beamrider curve is averaged over 2 runs rather than 3.

# B  Instructions Provided to Contractors

## B.1  MuJoCo

**Giving feedback**

Sign up for a slot in the spreadsheet. Then go to the appropriate URL's that we give you, and you'll be repeatedly presented with two video clips of an AI controlling a virtual robot.

**Look at the clips and select the one in which better things happen.** Only decide on events you actually witness in the clip.

**Here's a guide on what constitutes good and bad behavior in each specific domain:**

- **Hopper**: the "center" of the robot is the joint closest to the pointy end. The first priority is for the center of the robot to move to the right (moving to the left is worse than not moving at all). If the two robots are roughly tied on this metric, then the tiebreaker is how high the center is.

- **Walker**: the "center" of the robot is the joint where the three limbs meet. The first priority is for the center of the robot to move to the right. If the two robots are roughly tied on this metric, then the tiebreaker is how high the center is.
- **Swimmer**: the "center" of the robot is the mark in the middle of its body. The center should move to the right as fast as possible.
- **Cheetah**: the robot should move to the right as fast as possible.
- **Ant**: the first priority is for the robot to be standing upright, and failing that for the center of the robot to be as high up as possible. If both robots are upright or neither is, the tie breaker is whichever one is moving faster to the right.
- **Reacher**: the green dot on the robot arm should be as close as possible to the red dot. Being near for a while and far for a while is worse than being at an intermediate distance for the entire clip.
- **Pendulum**: the pendulum should be pointing approximately up. There will be a lot of ties where the pendulum has fallen and a lot of "can't tells" where it is off the side of the screen. If you can see one pendulum and it hasn't fallen down, that's better than being unable to see the other pendulum.
- **Double-pendulum**: both pendulums should be pointing approximately up (if they fall down, the cart should try to swing them back up) and the cart should be near the center of the track. Being high for a while and low for a while is worse than being at an intermediate distance the entire time.

If both clips look about the same to you, then click "tie". If you don't understand what's going on in the clip or find it hard to evaluate, then click "can't tell".

**You can speed up your feedback by using the arrow keys**
`left` and `right` select clips, up is a tie, down is "can't tell".

### FAQ

**I got an error saying that we're out of clips. What's up?** Occasionally the server may run out of clips to give you, and you'll see an error message. This is normal, just wait a minute and refresh the page. If you don't get clips for more than a couple minutes, please ping @tom on slack.

**Do I need to start right at the time listed in the spreadsheet?** Starting 10 minutes before or after the listed time is fine.

### B.2  Atari

*In this task you'll be trying to teach an AI to play Atari games by giving it feedback on how well it is playing.*

#### IMPORTANT. First play the game yourself for 5 minutes

Before providing feedback to the AI, play the game yourself for a five minutes to get a sense of how it works. It's often hard to tell what the game is about just by looking at short clips, especially if you've never played it before.

Play the game online for 5 minutes.[9] You'll need to press F12 or click the GAME RESET button to start the game. Then set a timer for 5 minutes and explore the game to see how it works.

#### Giving feedback

Sign up for a slot in the spreadsheet. Then go to the appropriate URL's that we give you, and you'll be repeatedly presented with two video clips of an AI playing the game.

**Look at the clips and select the one in which better things happen.** For example, if the left clip shows the AI shooting an enemy ship while the right clip shows it being shot by an enemy ship, then better things happen in the left clip and thus the left clip is better. Only decide on actions you actually witness in the clip.

**Here's a guide on what constitutes good and bad play in each specific game:**

- **BeamRider**: shoot enemy ships (good), and don't get shot (very bad)
- **Breakout**: hit the ball with the paddle, break the colored blocks, and don't let the ball fall off the bottom of the screen
- **Enduro**: pass as many cars as you can, and don't get passed by cars
- **Pong**: knock the ball past the opponent's orange paddle on the left (good), and don't let it go past your green paddle on the right (bad)
- **Qbert**: change the color of as many blocks as you can (good), but don't jump off the side or run into enemies (very bad)
- **SpaceInvaders**: shoot enemy ships (good), and don't let your ship (the one at the bottom of the screen) get shot (very bad)
- **SeaQuest**: Shoot the fish and enemy submarines (good) and pick up the scuba divers. Don't let your submarine run out of air or get hit by a fish or torpedo (very bad)
- **Enduro (even mode)**: Avoid passing cars OR getting passed by them, you want to stay even with other cars (not having any around is OK too)

**Don't worry about how the agent got into the situation it is in** (for instance, it doesn't matter if one agent has more lives, or is now on a more advanced level); just focus on what happens in the clip itself.

If both clips look about the same to you, then click "tie". If you don't understand what's going on in the clip or find it hard to evaluate, then click "can't tell". Try to minimize responding "can't tell" unless you truly are confused.

**You can speed up your feedback by using the arrow keys**
`left` and `right` select clips, `up` is a tie, `down` is "can't tell".

### FAQ

**I got an error saying that we're out of clips. What's up?** Occasionally the server may run out of clips to give you, and you'll see an error message. This is normal, just wait a minute and refresh the page. If you don't get clips for more than a couple minutes, please ping @tom on slack.

**If the agent is already dead when the clip starts, how should I compare it?** If the clip is after getting killed (but not showing the dying), then its performance during the clip is neither good nor bad. You can treat it as purely average play. If you see it die, or it's possible that it contains a frame of it dying, then it's definitely bad.

**Do I need to start right at the time listed in the spreadsheet?** Starting 30 minutes before or after the listed time is fine.