[Reviews · NeurIPS 2017]

Reviewer 1



The paper introduces a technique for reinforcement learning in scenarios in which the reward function is unclear by asking human experts for preferences between two trajectories. These preferences are used to inform a learned reward function, which in turn is used to improve the policy producing the trajectories. The paper produces little theory and no guarantees, but the approach is shown to be competitive with learning from the true reward function, with relatively little human interaction. The paper should be accepted because it is an important problem, an interesting approach, and the experiments are well-conducted and the results promising; I believe it will get a lot of attention from the community. However, there are many unanswered questions that I am curious about. First, how closely does the learned reward \hat r match the true reward function? I suspect human preferences are informed not strictly by reward, but rather by value, or something like it - so a kind of shaping reward might be appearing in cases where queries outperformed learning from the true reward function. The question of how good is your reward model seems important for extensions to other domains. Second, I'm unclear how the compared trajectories are created. Are the policies stochastic, so that trajectories diverge even as the policy stays the same, or are the trajectories generated from different starting states?

Reviewer 2



This paper implements previously existing systems for learning RL policies from human preferences in much higher dimensional problems. Learning from human preferences has be proposed before, in this special case the authors use comparisons between trajectories has the information about preferences. The method is a trivially extension without any major difficulties to apply previous preference elicitation methods using policy optimization methods.

Reviewer 3



Overall I find this paper is generally interesting, clearly presented, and technically sound. My concerns are that the contributions of this paper seems rather incremental when compared to previous work. Also, some of the experiments would benefit from further analysis. Let me elaborate below: Summary: This paper advocates a preference-based approach for teaching an RL agent to perform a task. A human observes two trajectories generated by different policies and indicates which one is better performing the desired task. Using these indications, the agent infers a reward signal that is hopefully consistent with the preferences of the human, and then uses RL to optimize a policy that maximizes the inferred rewards. My main hesitation for accepting this paper is that the method presented is not sufficiently different from previous work on preference-based RL (such as Wirth '16). The authors acknowledge that the main contribution of this work is to scale up preference-based RL to more complex domains by using deep reinforcement learning. I am less enthusiastic about this paper if the main algorithmic contribution is just using a NN to approximate the policy and extending to Mujoco/Atari. Finally, I am concerned with the fact that synthetic queries are outperforming RL using the true reward function. It seems to me that the synthetic query examples should do strictly worse than having the actual reward function as they are just eliciting preferences based on sub-trajectories of the real rewards. The fact that they are in some cases solidly outperforming standard RL is highly unexpected and, in my opinion, deserves a more rigorous analysis. I don't find the authors explanation of the learned reward function being smoother a satisfying one.